# Uncemented Tibial Fixation Has Comparable Prognostic Outcomes and Safety Versus Cemented Fixation in Cruciate-Retaining Total Knee Arthroplasty: A Meta-Analysis of Randomized Controlled Trials

**DOI:** 10.3390/jcm12051961

**Published:** 2023-03-01

**Authors:** Kai Chen, Jintao Xu, Hanhao Dai, Yaohui Yu, Yishu Wang, Yi Zhu, Tianqi Tao, Yiqiu Jiang

**Affiliations:** 1Nanjing First Hospital, Nanjing Medical University, 68 Changle Road, Nanjing 210029, China; 2Shengli Clinical Medical College of Fujian Medical University, Fuzhou 350000, China

**Keywords:** arthroplasty, uncemented tibial fixation, randomized controlled trial, knee replacement, osteoarthritis

## Abstract

Background: Cemented and uncemented fixation are the primary methods of tibial prosthesis fixation in total knee arthroplasty. However, the optimal fixation method remains controversial. This article explored whether uncemented tibial fixation has better clinical and radiological outcomes, fewer complications, and revision rates compared to cemented tibial fixation. Methods: We searched the PubMed, Embase, Cochrane Library, and Web of Science databases up to September 2022 to identify randomized controlled trials (RCTs) that compared uncemented total knee arthroplasty (TKA) and cemented TKA. The outcome assessment consisted of clinical and radiological outcomes, complications (aseptic loosening, infection, and thrombosis), and revision rate. Subgroup analysis was used to explore the effects of different fixation methods on knee scores in younger patients. Results: Nine RCTs were finally analyzed with 686 uncemented knees and 678 cemented knees. The mean follow-up time was 12.6 years. The pooled data revealed significant advantages of uncemented fixations over cemented fixations in terms of the Knee Society Knee Score (KSKS) (*p* = 0.01) and the Knee Society Score–Pain (KSS–Pain) (*p* = 0.02). Cemented fixations showed significant advantages in maximum total point motion (MTPM) (*p* < 0.0001). There was no significant difference between uncemented fixation and cemented fixation regarding functional outcomes, range of motion, complications, and revision rates. When comparing among young people (<65 years), the differences in KSKS became statistically insignificant. No significant difference was shown in aseptic loosening and the revision rate among young patients. Conclusions: The current evidence shows better knee score, less pain, comparable complications and revision rates for uncemented tibial prosthesis fixation, compared to cemented, in cruciate-retaining total knee arthroplasty.

## 1. Introduction

Osteoarthritis (OA) is the most common musculoskeletal disease worldwide and the primary cause of mobility loss, which has a detrimental effect on quality of life and increases healthcare costs [1]. Total knee arthroplasty has been widely recognized as the golden standard for end-stage osteoarthritis [2]. However, younger patients are undergoing elective knee replacement in higher numbers, and there is a greater focus on improving functional recovery and quality of life after TKA [3,4].

Currently, cemented fixation is the mainstream method for TKA. In the United States, the United Kingdom, and in other countries, more than 80% of TKA are performed in a cemented technique [5,6]. Advantages such as immediate fixation, inaccurate-cut compensation, and local antibiotic delivery make cemented fixation the gold standard for TKA, not to mention its excellent survival rate and prognosis [7]. Nevertheless, with the increasingly sophisticated surgical procedures, the increasing proportion of young people under 65 years of age, and the patients’ expectation of a better prognosis, problems such as osteolysis and aseptic loosening at the bone–cement interface have raised concerns about the long-term effectiveness of cemented TKA [8,9].

An alternative to cemented fixation called uncemented fixation, also known as “biologic fixation”, has emerged. Because there is no cement used, uncemented TKA theoretically has benefits such as preserving bone stock, avoiding cement fragmentation, and reducing the risk of implant loosening, contributing to more durable biological fixation of the prosthesis compared to conventional cemented TKA [10]. Uncemented TKA is becoming increasingly popular. For instance, 10% of the TKA in the US employ an uncemented technique in 2019 [5]. 

However, despite the theoretical advantages over the cemented technique, many studies have also reported that uncemented fixation showed no advantage and was even inferior to the traditional cemented technique in terms of cost and aseptic loosening due to the early imperfect techniques [11,12,13,14]. Moreover, aseptic loosening is one of the most common causes of revisions, especially when involving tibial prostheses [5,15,16]. However, with the development of advanced techniques such as the application of 3D printing technology and various types of coatings, uncemented fixation has been reported to have comparable results with cemented fixation in recent years [17,18,19]. 

Several reasons contribute to the increase in utilization of total knee replacement among younger patients, such as the increase in population size, the obesity epidemic, the growing prevalence of sports-related knee injuries, and expanded indications for total knee replacement [3]. Moreover, there is more stress stimulus on the implant in the younger population because they tend to engage in frequent intense physical activity, which will result in a greater likelihood of revision surgery [20]. The lifetime risk of revision (LTRR) is up to 35% for patients aged 50–54 years [21]. Thus, the risk of TKA revisions will be a factor that surgeons must consider when choosing a prosthetic fixation method in the future. Some researchers believe that uncemented fixation is the optimal choice for younger patients considering its ease of revision probability and better biological fixation.

Collectively, the option of cemented versus uncemented tibial prosthesis is still in dispute. Therefore, we performed a meta-analysis from published RCTs over 10 years to evaluate the optimal fixation approach in TKA. The purpose of this meta-analysis was to evaluate the following: (1) functional outcomes (Knee Society Knee Score [KSKS], Knee Society Function Score [KSFS], Knee Society Score–Pain [KSS–Pain], range of motion [ROM]); (2) radiology (Radiolucent line [RLL], maximum total point motion [MTPM]); (3) complications; and (4) revisions.

## 2. Materials and Methods

This research has been reported in line with PRISMA (Preferred Reporting Items for Systematic Reviews and Meta-Analyses).

### 2.1. Literature Search Strategy

A comprehensive literature search was conducted in September 2022 by two independent investigators (Chen K. and Xu J.). The primary sources were the electronic databases of PubMed, Embase, Cochrane, and Web of Science. The search keywords were centered on the terms “total knee arthroplasty”, “knee arthroplasty”, “cemented”, “uncemented OR cementless”, and “randomized controlled trial”. This process was performed iteratively until no additional articles could be identified. There were no restrictions on the article type or language of publication, except for restrictions on the date of publication.

### 2.2. Inclusion and Exclusion Criteria

Trials were selected based on the following inclusion criteria: (1) population: patients undergoing primary total knee arthroplasty; (2) intervention: uncemented tibial fixation in total knee arthroplasty; (3) comparison: cemented tibial fixation in total knee arthroplasty; (4) outcome measures—at least one of the following outcome measures was reported: functional outcomes, radiological evaluation, complication, revision; and (5) study design: only RCTs. Exclusion criteria were as follows: (1) randomized trials without a placebo or treatment group; (2) articles without available outcome data; (3) published before 2010; (4) duplicate reports; (5) reviews or case reports.

### 2.3. Quality Assessment

The quality of the included RCTs was assessed by two independent researchers based on the Cochrane Risk of Bias Tool for randomized controlled trials [22], and each quality item was graded as low risk, high risk, or unclear risk. The quality of each study was assessed using the following 7 items: randomization sequence generation, allocation concealment, blinding of participants and personnel, blinding of outcome assessment, incomplete outcome data, selective reporting, and other biases. Each criterion could be further graded as low, high, or unclear risk. 

### 2.4. Data Extraction

Two investigators (Chen K. and Dai H.) independently extracted the following data from each included study: first author name, year of publication, country of origin, study design, sample size, mean age, sex ratio, intervention measures, follow-up time, and outcome measures. Disagreements between authors were resolved by a third reviewer (Jiang Y.). We contacted the corresponding authors in case of incomplete data.

### 2.5. Statistical Analysis

Statistical analyses were conducted using Review Manager Software (RevMan Version 5.4; The Cochrane Collaboration, Copenhagen, Denmark). Differences were expressed as relative risks (RRs) with 95% confidence intervals (CIs) for dichotomous outcomes, and mean differences (MD) with 95% CIs for continuous outcomes. A *p*-value of <0.05 was the threshold for statistical significance. Statistical heterogeneity between summary data was evaluated using the *I*^2^ statistic. When there was no statistical evidence of heterogeneity (*I*^2^ < 50%, *p* > 0.1), a fixed effects model was adopted; otherwise, a random effects model was chosen. The publication bias was assessed using a funnel plots diagram. Sensitivity analyses were undertaken to determine the potential source of heterogeneity when significant. All figures were automatically generated via RevMan except for Figure 1.

## 3. Results

### 3.1. Study Identification and Characteristics

A total of 394 studies were identified from four databases (PubMed, Embase, WOS, Cochrane), of which 166 were duplicate studies that were removed. Titles and abstracts of the remaining 226 studies were screened, and then 41 of them were subjected to full-text review. Finally, nine studies that fulfilled the predetermined inclusion criteria were included in this meta-analysis [23,24,25,26,27,28,29,30,31] (Figure 1). The mean follow-up time was 9.0 years, ranging from 2.0 to 23.8 years. Characteristics of included studies are displayed in Table 1. The risk bias assessment of each study is displayed in Figure 2, with a detailed traffic light plot. Funnel plots were drawn using RevMan to assess the reporting bias for each outcome, and figures are included in Appendix A Appendix A.

### 3.2. Clinical Outcomes

Eight studies, including 603 uncemented and 595 cemented knees, showed the benefit of uncemented fixation compared to cemented fixation in the KSKS (MD: 0.98; 95% CI: 0.20 to 1.76; *p* = 0.01; *I*^2^ = 30%; Figure 3a). Five studies, including 248 cemented and 242 uncemented knees, also showed the benefit of uncemented fixation compared to cemented fixation in the KSFS (MD: 1.57; 95% CI: −1.17 to 4.31; *p* = 0.26; *I*^2^ = 0%; Figure 3b). However, these differences were not statistically significant. Three studies showed that uncemented fixation seemed to have better outcomes in terms of the KSS–Pain (MD: 2.79; 95% CI: 0.44 to 5.14; *p* = 0.02; *I*^2^ = 0%; Figure 3c). Eight studies that reported on ROM indicated a slight advantage of uncemented fixation over cemented fixation (MD: 1.15; 95% CI: −0.04 to 2.33; *p* = 0.06; *I*^2^ = 10%; Figure 3d) that were not statistically significant. After a sensitivity analysis, it was found that with the exclusion of Kim’s study, no significant difference was found in terms of the KSKS (MD: 0.95; 95% CI: −0.43 to 2.32; *p* = 0.13; *I*^2^ = 40%). The mean age in three studies was younger than 55 years old [27,28,31], and in one additional study, it was younger than 60 years old [26]. When compared with younger patients in these four studies, no significant differences were found in clinical outcomes. Overall, based on the available evidence, it was revealed that uncemented tibial fixation was superior to cemented tibial fixation in the KSKS and KSS–Pain, but there was no significant difference concerning functional outcomes.

### 3.3. Radiological Outcomes

Six studies with 541 cemented and 540 uncemented knees reported data for RLL. The pooled results did not demonstrate significant differences between the cemented and uncemented groups (RR: 0.97; 95% CI: 0.62 to 1.53; *p* = 0.90; *I*^2^ = 36%; Figure 4a). This result, based on a fixed-effects model, was unchanged by the sensitivity analysis. Three studies, including 82 cemented and 84 uncemented knees, indicated that cemented fixations had a smaller MTPM than uncemented fixations (MD: 0.33; 95% CI: 0.08 to 0.58; *p* = 0.01; *I*^2^ = 90%; Figure 4b). However, all three studies describing MTPM showed that the uncemented group showed more MTPM occurring only in the first three months after prosthesis implantation, whereas when MTPM in the third month after prosthesis implantation was used as the baseline, there was no significant difference in MTPM generated by the two modalities. Uncemented fixations usually take longer to achieve biological fixation, whereas cemented fixations can achieve good stability in a short period of time, which also explains the difference between long-term migration and initial migration.

### 3.4. Complications

A total of seven studies provided data on aseptic loosening, which contained 618 uncemented and 617 cemented knees. Two studies were not able to be analyzed because there were no aseptic loosening failures included. The pooled results did not demonstrate significant differences between the uncemented and cemented groups (RR: 1.12; 95% CI: 0.44 to 2.88; *p* = 0.79; *I*^2^ = 0%; Figure 5a). There was no significant heterogeneity among the studies, and a fixed effects model was used. Seven studies reported data for infection with 624 uncemented and 623 cemented knees. No significant difference was identified in the infection rate (RR: 0.86; 95% CI: 0.30 to 2.44; *p* = 0.77; *I*^2^ = 0%; Figure 5b). The incidence of thrombosis was reported in two studies, and no significant difference was found (RR: 0.66; 95% CI: 0.13 to 3.32; *p* = 0.62; *I*^2^ = 0%; Figure 5c). These results, based on a fixed-effects model, were unchanged by the sensitivity analysis assumptions. No significant difference was found among studies with younger patients (RR: 0.97; 95% CI: 0.32 to 2.90; *p* = 0.95; *I*^2^ = 13%) in terms of aseptic loosening. As a result of these findings, it indicated that uncemented fixation had a low complication rate comparable to that of cemented fixation.

### 3.5. Revisions

Eight included studies reported the occurrence of revisions, of which one could not be analyzed because no revision case was reported. The pooled results showed that there was no significant difference between uncemented tibial fixations and cemented fixations in terms of the revision rate (RR: 1.02; 95% CI: 0.55 to 1.89; *p* = 0.95; *I*^2^ = 0%; Figure 6a). Furthermore, to determine whether there is a difference in survivorship between these two fixations in young patients (younger than 60 years old), four studies were analyzed; no significant difference was found (RR: 0.96; 95% CI: 0.49 to 1.88; *p* = 0.91; *I*^2^ = 0%; Figure 6b). Subgroup analysis based on follow-up time revealed no significant difference in revision rates, either short-term, mid-term, or long-term (Figure 7).

## 4. Discussion

The most important finding in this meta-analysis was that uncemented tibial fixation had a significant advantage over cemented tibial fixation in terms of clinical outcomes, including the KSKS and KSS–pain. The significant difference between the KSKS disappeared after sensitivity analysis, while in KSS–pain it did not. In terms of radiological results, no significant differences were shown in terms of RLL. However, for MTPM, the motion of the cemented group was significantly smaller than that of the uncemented group. There were no significant differences in postoperative complications, including aseptic loosening, infection and thrombosis, and revision rates between the two fixation methods, and these two fixation methods neither showed significant differences in the short or long term. It was also interesting to find that the mean BMI of all included studies was above 25. This may be due to the fact that obesity is one of the major risk factors for osteoarthritis.

Uncemented fixation uses a ‘biological fixation’, which means that there is a direct structural and functional connection between ordered, living bone and the surface of a load-carrying implant [32]. With a porous-coated surface, uncemented fixation theoretically allows for better clinical outcomes by avoiding the adverse effects of cement fragments, encouraging new bone growth, contributing to the long-term survival of implants, and thus creating a long-lasting stable biological fixation [33,34]. The surface of the implant plays an important role in direct skeletal fixation, and porous implants are very effective for skeletal fixation. Without the use of bone cement, uncemented prostheses avoid changes in the bone/cement interface that may occur over time after prosthesis implantation, which may lead to osteolysis. This implies that for the younger population with a higher bone density, there should be a greater advantage in terms of joint stability and functional recovery from uncemented joints. No significant differences were found between the uncemented and cemented fixations regarding RLL as well as complications. This may be due to advances in cementless fixation techniques. In the early period, uncemented fixation was often associated with more implant failures and complications due to immature tibial prosthesis locking mechanisms and fixation devices [35,36,37]. Previous uncemented prostheses often used mechanical devices, including screws, to obtain fixation; however, this can lead to the formation of bone fragments, and subsequently increase the risk of osteolysis. With the porous or rough surface design of the prosthesis, and the application of various types of coatings, no screws are required to achieve a more stable mechanical interlocking, limiting the micromotion of the prosthesis and resulting in the number of RLL and complications comparable to conventional cemented fixation [10,38]. It is also possible that the cemented fixation, as a proven type, has inherently low adverse outcomes. The number of knees in the RCTs included was not sufficient to reflect the difference, resulting in no significant difference in the overall incidence between the two groups. This needs to be corroborated by more high-quality RCTs with longer follow-up times and strict inclusion criteria. In addition, uncemented fixation requires more from the operator to make a perfect fit of the prosthesis, which may be another reason for why the current results do not fully reflect the theoretical advantages of the uncemented fixation. The difference in MTPM was mainly seen within the first 3 months after tibial prosthesis implantation, mainly because the bone cement could fill the bone cutting defect at an early stage. However, no significant differences were found in MTPM between the two fixation modalities after 3 months, and there was no statistical difference in the number of RLLs, showing good prosthetic stability with uncemented fixation. This may perhaps have been due to the formation of a stable and tight biological structure between the prosthesis/bone interface in the uncemented group.

In some recent studies, it was concluded that there was a statistical difference between uncemented and cemented fixations in young patients in terms of clinical outcomes and complications [39,40]. However, with unclear inclusion criteria and a large timespan of published studies, it may be assumed that the conclusions are not reliable and require more rigorous evidence. According to studies without the age limit, the clinical outcomes, including the KSKS and KSS–Pain, suggested an advantage of uncemented tibial fixations. However, after the extraction of one study, our results of clinical outcomes changed. Considering that Kim’s study had a follow-up time of more than 23 years, accounting for 67.8% of the results, and the prosthesis was not of a newer generation, this may explain why the results were not stable; hence, more studies with consistent prostheses are needed. RLL is the radiolucent interval between the prosthesis and bone and is widely considered to be closely associated with aseptic loosening. However, it has also been reported that the clinical significance of RLLs is not well defined and their incidence is low, and no correlation has been found between their presence and clinical loosening [41]. Hampton et al. indicated that uncemented knees with tantalum metal components had better radiological analysis compared with those of the cemented group [42]. Rand et al. found that the numbers of RLLs occurring after the two fixations were similar [43]. This difference in RLL outcomes may account for the difference in uncemented prostheses used at different times. We included the most recent studies from the last decade, but it is still hard to ensure that all of the prostheses were up to date. The present meta-analysis showed no significant difference in complications and the revisions rate between uncemented and cemented fixations. The most common reasons for knee revision surgery were infection and inflammatory reaction at 24.8%, mechanical loosening at 24.2%, and other mechanical complications at 20.7% [5] (American Joint Replacement Registry (AJRR): 2020 Annual Report. Rosemont, IL: American Academy of Orthopedic Surgeons (AAOS), 2020) (“American Joint Replacement Registry (AJRR): 2020 Annual Report. Rosemont, IL: American Academy of Orthopedic Surgeons (AAOS), 2020,”) [5]. In our meta-analysis, nine studies reported aseptic loosening, seven studies reported infection, and two studies reported thrombosis. Based on our study, the complication rates of uncemented and cemented fixations were comparable and low, which is consistent with the findings of several previous studies [39,40,44]. Especially in terms of aseptic loosening in young patients, uncemented fixation modalities showed comparable results to gold-standard cemented fixation in studies of the last decade.

Patients included in this study were treated with Posterior-Stabilized TKA. Preservation or reconstruction of the posterior cruciate ligament remains as one of the issues facing TKA, and studies over the last decade have shown that PS or CR TKA remains controversial in terms of joint stability and functional recovery. Proponents of CR believe that preservation of the posterior cruciate ligament maximizes the preservation of knee physiology and provides advantages in terms of local sensation, balance, and kinematics. In contrast, proponents of PS argue that PS increases ROM as well as or better than femoral rollback. In terms of revision rates, a study by Peter et al. found higher revision rates in CR than PS TKA in exostosis-type osteoarthritis, possibly because the nonfunctional PCL was preserved, even though this may yield slightly better clinical outcomes [45]. However, a study by Anneke et al. confirmed that the PS type had a higher rate of mid-term revision, which may be due to a higher rate of loosening caused by the higher shear force of the tibial prosthesis in PS TKA [46]. Similarly, Meredith et al. found that coronal relaxation was significantly increased in PS TKA and resulted in higher polyethylene wear, but these differences did not appear to have an impact on clinical outcomes and survival rates.

To determine the optimal approach for fixation of the tibial prosthesis, our study included nine high-quality randomized controlled studies in the last ten years for the first time, providing a higher quality evidence base than previous studies that included fewer retrospective studies. Moreover, we further subdivided various evaluation criteria, including the KSS as well as radiological outcomes, complications and revisions, providing a more detailed basis for the choice of the tibial fixation method.

Despite the strengths of our research, there are still some limitations. Firstly, since the inception of total knee arthroplasty, there have been significant differences between the design, materials, and use of different prostheses, even when the same uncemented prosthesis has a metallic microporous or nanoparticle design on the surface, in addition to differences in whether patellar replacement was performed intraoperatively, and in the selection of femoral prostheses. When analyzing the KSKS, sensitivity analysis revealed that the difference was no longer significant when Kim’s 2020 study was removed. However, after reviewing the design, procedures, and results of Kim’s study, we finally decided to retain the results including Kim. Initially, no significant study heterogeneity that could lead to changes in the results was found in his study, and the prosthesis used in Kim’s study is still widely used, making his conclusions more in line with the status quo. However, there is no denying that even though we minimized bias by limiting the time of publication of the article, these differences may still affect our conclusions; therefore, there is an urgent need for larger randomized controlled trials in which only the variable of bone cement is present. Secondly, even though nine full RCTs were included that included 917 patients, the number of included studies still needs to be improved. TKA is a very well-established procedure, with a low rate of postoperative complications. In several of the included studies, the complication rate was almost zero. It was difficult to find differences on an order of magnitude in this study. Thirdly, the follow-up period of our included studies ranged from 2 to 23.8 years, and some of the included studies were age-restricted; the relationship between age and tibial BMD may have led to biased results. A meta-analysis stratified by age and duration of follow-up could address this issue. Moreover, the available clinical results only considered the reliability of different prosthetic designs at the time of primary TKA, and did not statistically or adequately take into account the benefit of bone cement selection or not for revision TKA. Considering that both modalities have low revision rates, the impact of the choice of initial TKA procedure at the time of revision TKA needs to be analyzed. Finally, the number of included studies was too small for some of the results to assess reporting bias that may have affected the quality of the evidence.

Based on the current evidence, we confirmed the advantage of uncemented fixation of the tibia in clinical recovery and pain relief. This is in line with the theoretical superiority of uncemented fixation. With continuous advancements in the field of auxiliary materials, material science, and system design, the complications as well as joint loosening have become comparable to conventional cemented fixation, and there are potential advantages in long-term stability. Although there is no difference in complications and revision rates, given the significantly higher risk of revision in young TKA patients, uncemented fixation remains the more desirable option, relying on its better clinical outcomes and lower disruptions of bone volume and biological fixation. For patients with a lower risk of revision, bone cement remains the preferred option as a proven, cost-effective, and safe modality.

## 5. Conclusions

Our meta-analysis of nine randomized controlled trials confirmed that cementless tibial prosthesis fixation has some advantages in KSKS and KSS–Pain and has comparable results with cemented tibial prosthesis fixation in KSFS, RLL, complications, and revision rates, but has some disadvantages in MTPM. However, this disadvantage may only exist in the short term after implant fixation. Therefore, based on the results of this study, a better knee score as well as less pain may be achieved with an uncemented tibial component for patients who are about to undergo TKA, regardless of cost. In terms of cost, some recent studies from the United States and the Netherlands reported that with increased use and industrial production, uncemented fixation is not more expensive than traditional cemented fixation in these countries [47,48,49]; however, this conclusion did not apply to all countries and regions, thus cemented fixation remains as an appropriate option with beneficial prospects. Moreover, a more standardized, easy-to-perform, and affordable joint replacement procedure is urgently required to reduce the impact of operator variability on patients’ prognoses.

## Figures and Tables

**Figure 1 jcm-12-01961-f001:**
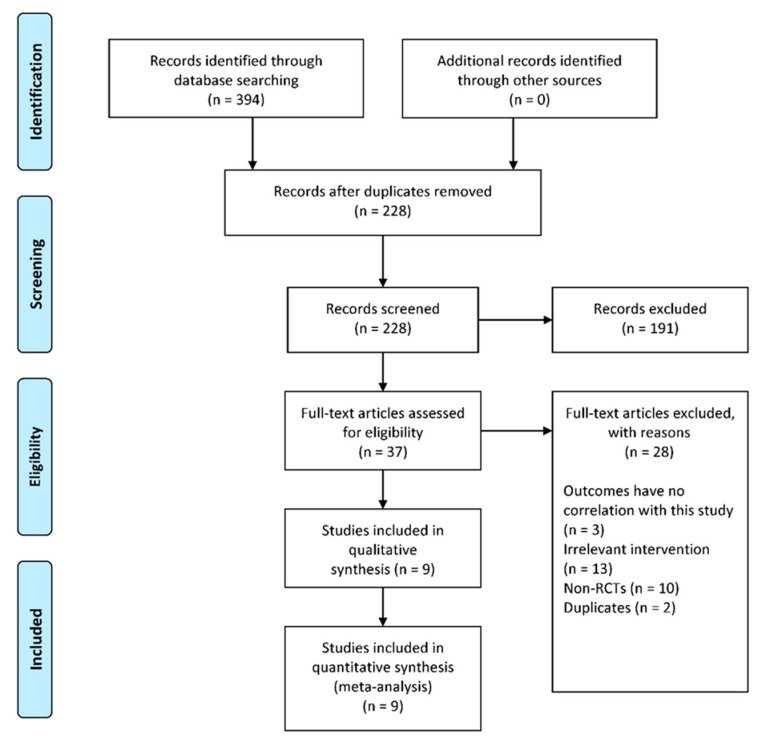
PRISMA flow diagram of study selection.

**Figure 2 jcm-12-01961-f002:**
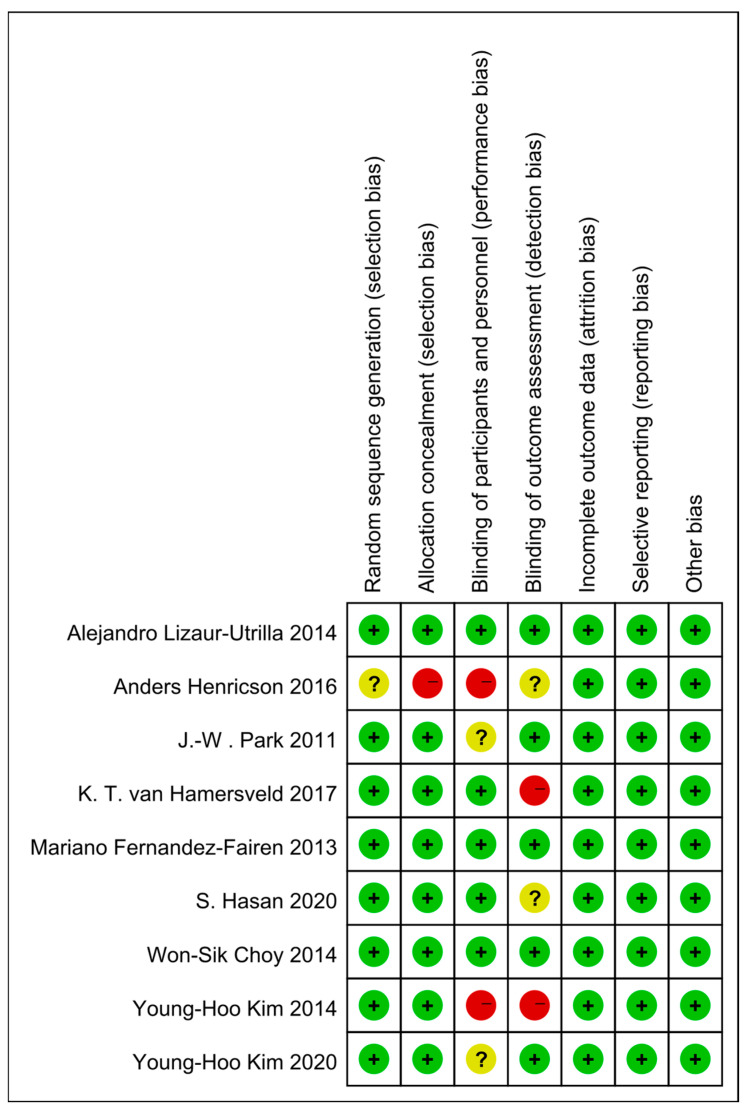
Risk of bias summary [23,24,25,26,27,28,29,30,31]. “+”: Low risk of bias; “−”: High risk of bias; “?”: Unclear risk of bias.

**Figure 3 jcm-12-01961-f003:**
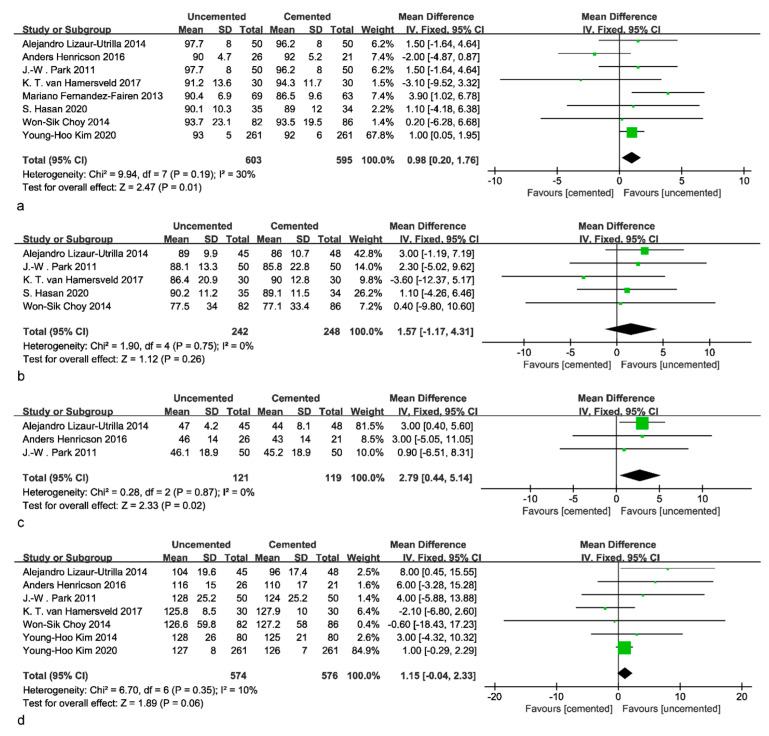
Forest plot of KSS and ROM. (**a**) Forest plot of KSKS [23,24,25,26,27,28,29,30]. (**b**) Forest plot of KSFS [23,25,28,29,30]. (**c**) Forest plot of KSS–Pain [26,28,29]. (**d**) Forest plot of ROM [23,26,27,28,29,30,31].

**Figure 4 jcm-12-01961-f004:**
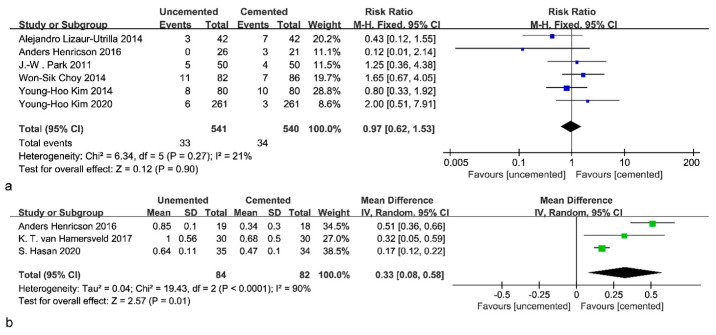
Forest plot of RLL and MTPM. (**a**) Forest plot of RLL [23,26,27,28,29,31]. (**b**) Forest plot of MTPM [25,26,30].

**Figure 5 jcm-12-01961-f005:**
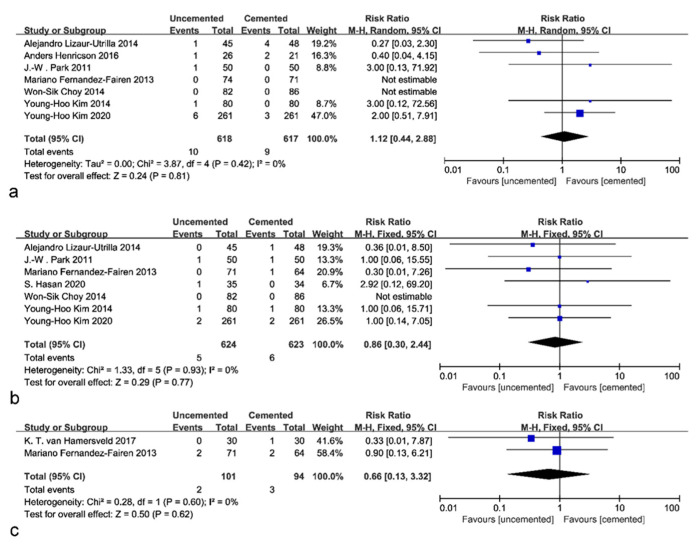
Forest plot of complications. (**a**) Forest plot of aseptic loosening [23,24,26,27,28,29,31]. (**b**) Forest plot of infections [23,24,25,27,28,29,31]. (**c**) Forest plot of thrombosis [24,30].

**Figure 6 jcm-12-01961-f006:**
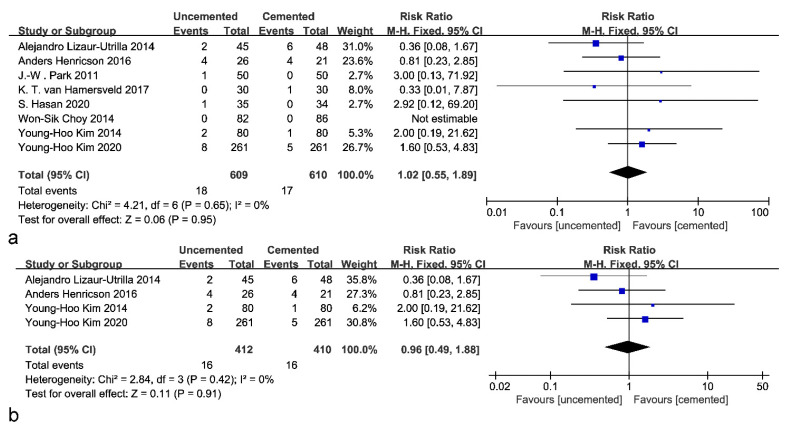
Forest plot of revisions. (**a**) Forest plot of revisions (without age limit) [23,25,26,27,28,29,30,31]. (**b**) Forest plot of revisions in young patients [26,27,28,31].

**Figure 7 jcm-12-01961-f007:**
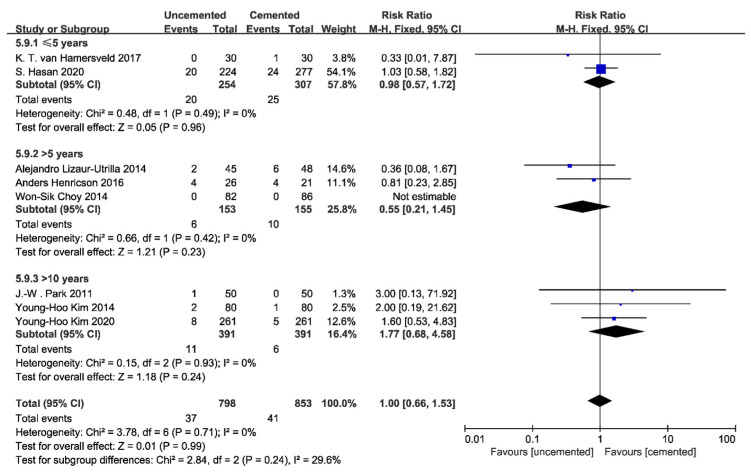
Forest plot of revisions by time [23,25,26,27,28,29,30,31].

**Table 1 jcm-12-01961-t001:** Characteristics of the studies included in the meta-analysis. Abbreviations: C: cemented; UC: uncemented; CR: cruciate-retaining; PS: posterior-stabilized; KSS: Knee Society Score; ROM: range of motion; RLL: radiolucent line; NA: not available.

Author	Year	Level of Evidence	No of Patients	No of Knees (C/UC)	Male (%)	Mean Age (C/UC)	CR or PS	Femoral Prosthesis	Tibial Prosthesis	Patellar Resurfacing	Mean Follow-Up (Years)	Outcomes
Hasan [25]	2020	I	69	34/35	52.17%	66/65	CR	Random	Random	No	2.0	Complications, Revision
Kim [27]	2020	I	261	261/261	31.03%	62.5	CR	Random	Random	Yes	23.8	KSS, ROM, RLL, Complications, Revision, Survivorship
Hamersveld [30]	2017	I	60	30/30	46.67%	65.7/66.8	CR	Random	Random	No	5.0	KSS, Complications, Revision
Henricson [26]	2016	I	33	26/21	56.10%	54	CR	NA	Random	No	10.0	KSS, ROM, RLL, Complications, Revision, Survivorship
Choy [23]	2014	I	126	86/82	7.94%	69/65	CR	UC	Random	No	9.5	KSS, ROM, RLL, Complications, Revision
Lizaur-Utrilla [28]	2014	I	93	48/45	27.96%	52/51.4	CR	UC	Random	NA	7.1	KSS, RLL, Complications, Revision, Survivorship
Kim [31]	2014	I	80	80/80	21.25%	54.3/54.3	CR	Random	Random	Yes	16.6	ROM, RLL, Complications, Revision
Fernandez-Fairen [24]	2013	I	145	71/74	24.83%	60/61	CR	UC	Random	No	5.0	KSS, RLL, Complications
Park [29]	2011	I	50	50/50	22.00%	58.4	CR	Random	Random	Yes	13.6	KSS, ROM, RLL, Complications, Revision, Survivorship

## Data Availability

No new data were created or analyzed in this study. Data sharing is not applicable to this article.

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
