# Peer review of "Uncemented Tibial Fixation Has Comparable Prognostic Outcomes and Safety Versus Cemented Fixation in Cruciate-Retaining Total Knee Arthroplasty: A Meta-Analysis of Randomized Controlled Trials"

_jcm, 2023, doi:10.3390/jcm12051961_

Round 1
Reviewer 1 Report
It is stated in a general way in the study that tibial fixation in knee prostheses is substantially equal in results for both cemented and uncemented. However, only CR prostheses are analyzed in the data while PS are completely absent. Considering that the stresses at the prosthesis-bone interface between these two types of implants are substantially different I would consider it important to emphasize the lack of validity of the claims reported by the authors about PS prostheses. I would, in addition, make a brief mention about the differences in mechanical stresses at the prosthesis-bone interface present between CR and PS prostheses. So I would also correct the title by specifying that the evaluation of the results is only about CR TKRs.
Author Response
Response to Reviewer 1 Comments
Thank you for your comments and suggestion concerning our manuscript. The comments and suggestions are all valuable and immensely helpful for revising and improving our paper, as well as the important guiding significance to our research. We have studied comments carefully and have made correction which we hope to meet with approval. Below we provide the point-by-point responses. All modifications in the manuscript have been highlighted in red.
Sincerely,
- Considering that the stresses at the prosthesis-bone interface between these two types of implants are substantially different I would consider it important to emphasize the lack of validity of the claims reported by the authors about PS prostheses. Make a brief mention about the differences in mechanical stresses at the prosthesis-bone interface present between CR and PS prostheses.
Response: Thank you for your kind suggestion. We have made a brief description in the discussion in the revised manuscript. The UK National Joint Registry 2020 data shows that CR is the most popular design philosophy (67.4%), followed by PS (23.1%), MB (6.9%), and the least common is MP (2.6%). This may partly explain the fact that all the studies we included were CRs. However, it is undeniable that there is also a considerable body of research confirming the superiority of PS in functional recovery and especially in ROM. Therefore, the lack of analysis and discussion of PS in our study has led to the loss of some clinical guidance for surgeons accustomed to PS.
- Correct the title by specifying that the evaluation of the results is only about CR TKRs.
Response: Thank you for your constructive suggestion. We have featured this in the title of the revised manuscript.
Reviewer 2 Report
For a very traditional topic, the authors did a meta-analysis of only RCTs.
In the subject of this study, it seems that clinicians are most curious about whether the results of uncemented tibial fixation are equivalent to cemented and what benefits are there. In this respect, it is regrettable that this study did not analyze such differences prominently.
Overall, English expressions and descriptions require revision.
Abstract
The abstract should summarize the subject of this study and its results.
Background and conclusion did not deliver accurate information to the readers. This part needs to be rewritten.
Introduction
The content of the first paragraph is too trite, and I think it would be good to delete it as too verbose before getting into the current topic.
Clinically, most cemented fixation still has many advantages, and it is being performed overwhelmingly. And its longevity has already been proven.
So why is uncemented fixation necessary? I think it is meaningful as a preparation for revision TKA that may be needed in the future in young patients. In this flow, it is considered that the introduction part should be improved and analyzed with more focus on this stream.
When writing the first 'TKA', write it in parentheses.
Methods
Why wasn't the Western Ontario and McMaster Universities Osteoarthritis Index (WOMAC) scores, University of California, Los Angeles (UCLA) activity score used as a standard?
17. Nam, D.; Lawrie, C.M.; Salih, R.; Nahhas, C.R.; Barrack, R.L.; Nunley, R.M. Cemented Versus Cementless Total Knee Arthro-336 plasty of the Same Modern Design: A Prospective, Randomized Trial. J Bone Joint Surg Am 2019, 101, 1185-1192, 337 doi:10.2106/JBJS.18.01162.
As with the above references cited by the authors, when making comparisons, the conditions need to be somewhat equivalent. Is age equally distributed in the population in the study analyzed by the authors?
Looking at the RCTs citated by the authors, it may be possible that these age factor are not well distributed.
In this topic, to claim that stability and longevity in uncement are equivalent compared to cemented, it is necessary to consider age, which can have a significant effect on the bone quality of the knee joint, and some of the references are analyzed for young patients, so There is a possibility of bias due to this.
Discussion
As pointed out earlier, the reason readers are interested in this comparative study is the question of whether this surgical method would be equivalent to that of conventional cemented in situations where uncemented should be used.
Therefore, a detailed discussion is needed for young patients or those who need to prepare for revision.
And in the case series of uncemented TKA, the design of the prosthesis used may be different from the prosthesis used as cemented in screw or stem. These areas also require descriptive discussions.
Limitation of this study should be added.
Conclusion
please rewrite to summary the results of this analysis.
Author Response
Response to Reviewer 2 Comments
Thank you for your comments and suggestion concerning our manuscript. The comments and suggestions are all valuable and extremely helpful for revising and improving our paper, as well as the important guiding significance to our research. We have studied comments carefully and have made correction which we hope to meet with approval. Below we provide the point-by-point responses. All modifications in the manuscript have been highlighted in red.
Sincerely,
- English expressions and descriptions require revision.
Response: Thank you for your kind suggestion. We revised the whole manuscript carefully to avoid language errors. In addition, we consulted a professional editing service from MDPI English Service and asked several colleagues who are native English speakers to check the English. We believe that the language is now acceptable for the review process.
- The abstract should summarize the subject of this study and its results.
Response: Thank you for your constructive suggestion. Our initial manuscript did lack a comprehensive and prominent description of the subject of this study and its results in the abstract section, which we have corrected and rewritten it in the revised manuscript.
- Background and conclusion did not deliver accurate information to the readers. This part needs to be rewritten.
Response: Thank you for your constructive suggestion. We rewrote the background and results in the abstract to try to make the description of the abstract more relevant. More content about the purpose of our study was added, and content not closely related to the core purpose of this study was removed.
- The content of the first paragraph is too trite, and I think it would be good to delete it as too verbose before getting into the current topic.
Response: Thank you for your constructive suggestion. We reviewed the first paragraph in the introduction and considered that it was indeed too trite, so we removed almost half of it and only gave a brief introduction to the important parts. We have removed the excessive introduction of TKA and focused on the way the prosthesis is fixed in TKA.
- So why is uncemented fixation necessary? I think it is meaningful as a preparation for revision TKA that may be needed in the future in young patients. In this flow, it is considered that the introduction part should be improved and analyzed with more focus on this stream.
Response: Thanks for your constructive suggestion. We did lack an explanation of the necessity of non-cemented fixation in our presentation, and we have added a relevant introduction.
- When writing the first 'TKA', write it in parentheses.
Response: Thanks for your comments. The discipline of the academic paper format is very important. We corrected the error at once and formalized the abbreviation.
- Why wasn't the Western Ontario and McMaster Universities Osteoarthritis Index (WOMAC) scores, University of California, Los Angeles (UCLA) activity score used as a standard?
Response: Thanks for your kind question. We also consider that the clinical outcome of TKA should be evaluated using a variety of objective evaluation criteria. However, out of a total of 9 included studies, only 5 reported WOMAC in clinical outcomes, one of which used Short-term WOMAC; therefore, we did not use WOMAC as one of the observations due to the lack of a sufficient number of studies. The University of California, Los Angeles (UCLA) activity score was not used for the same reason. Therefore, we used the American Knee Society Score as the primary clinical outcome evaluation criterion, which is also a widely used knee evaluation measure.
- As with the above references cited by the authors, when making comparisons, the conditions need to be somewhat equivalent. Is age equally distributed in the population in the study analyzed by the authors? Looking at the RCTs citated by the authors, it may be possible that these age factors are not well distributed.
Response: Thank you for your constructive question. The mean age of both UC and C groups was reported in 6 of the 9 RCTs we included, and the complete randomization grouping method was not reported in only 1 of these 9 studies, but in the remaining 8, as reflected in Figure 2. risk of bias summary. Randomization would ensure that potential confounding variables, including age, are equally distributed between the two groups, and therefore, we believe that the age distribution can be considered comparable between the two groups. However, some of these studies describe age only in terms of mean and extreme values, which blurs the distribution of age and may indeed lead to the problem of an unequal distribution of age with small probability.
- In this topic, to claim that stability and longevity in uncemented are equivalent compared to cemented, it is necessary to consider age, which can have a significant effect on the bone quality of the knee joint, and some of the references are analyzed for young patients, so There is a possibility of bias due to this.
Response: Thank you for your kind suggestion. We added this section to our discussion considering the age span and the fact that some of the studies restricted age, which was one of the issues that contributed to the limitations of our study and could be a confounding factor causing bias.
- A detailed discussion is needed for young patients or those who need to prepare for revision.
Response: Thanks for your constructive suggestion. Since younger patients have higher activity levels and a higher lifetime risk of revision (LTRR), selection of tibial prosthesis becomes even more important. Uncemented fixation is an ideal choice due to the absence of bone cement, as well as better long-term clinical outcomes and a lower risk of aseptic loosening. We have discussed this section further in the revised manuscript.
- And in the case series of uncemented TKA, the design of the prosthesis used may be different from the prosthesis used as cemented in screw or stem. These areas also require descriptive discussions.
Response: Thanks for your constructive suggestion. Different prosthesis designs can lead to potential selection bias. The elastic modulus of metal blocks and cement differ, which can result in different stress distributions in the proximal tibia. For different sizes of tibial defects, for example, the metal block and cement-screw techniques also have their own advantages [Liu, Y.; Zhang, A.; Wang, C.; Yin, W.; Wu, N.; Chen, H.; Chen, B.; Han, Q.; Wang, J. Biomechanical comparison between metal block and cement-screw techniques for the treatment of tibial bone defects in total knee arthroplasty based on finite element analysis. Computers in Biology and Medicine 2020, 125, 104006, doi:https://doi.org/10.1016/j.compbiomed.2020.104006]. We have added a description of this section to the revised manuscript.
- Limitation of this study should be added.
Response: Thank you for your constructive advice. After careful consideration of your comments on the paper, we recognize that our study needs to elucidate more of its limitations. We have added some limitations to the discussion, including selection bias due to confounding factors such as age limitations, prosthesis design and treatment of the femoral prosthesis and patella in TKA. We hope that our rewrite will make the limitations of this paper as clear as possible and provide some reference for subsequent studies.
- Conclusion, please rewrite to summary the results of this analysis.
Response: Thank you for your constructive advice. We have revised the summary to clarify the purpose of this study, the results, and its guidance for clinical selection.
Reviewer 3 Report
The study investigated the differences in outcomes between cemented and uncemented tibial component in knee prosthesis. It is a meta-analysis of RCTs.
The topic of the study is very interesting. The quality of the included studies is consistently high. In my opinion, the study is methodologically well conducted. Furthermore, the article is well written in all its parts: the introduction is adequate, the methods clear and reproducible, the statistical analysis well described, the results linear, the discussion sufficiently detailed and the conclusions consistent with the results.
I only have 2 considerations for the authors:
1) please include funnel plots for the analysis of publication bias for each outcome examined in the meta-analysis (possibly as supplementary material);
2) in my opinion the aspect of functional recovery and post-operative rehabilitation needs to be discussed in more detail: please describe in the results and comment in discussion what you have found in the various RCTs included.
Thank you.
Author Response
Response to Reviewer 3 Comments
Thank you for your comments and suggestion concerning our manuscript. The comments and suggestions are all valuable and extremely helpful for revising and improving our paper, as well as the important guiding significance to our research. We have studied comments carefully and have made correction which we hope to meet with approval. Below we provide the point-by-point responses. All modifications in the manuscript have been highlighted in red.
Sincerely,
- Please include funnel plots for the analysis of publication bias for each outcome examined in the meta-analysis (possibly as supplementary material)
Response: Thank you for your constructive advice. We used RevMan to plot funnel plots of all study results, some of which may lack statistical significance due to the small number of included studies, but most of which show a low publication bias for our analysis. All figures are available in the supplementary material (FigureS1-12).
- In my opinion the aspect of functional recovery and post-operative rehabilitation needs to be discussed in more detail: please describe in the results and comment in discussion what you have found in the various RCTs included.
Response: Thank you for your constructive suggestion. We have added to the discussion the significance of different tibial prosthesis fixation modalities for functional recovery and postoperative rehabilitation in young patients. Since younger patients have higher activity levels and thus tend to have a significantly higher chance of revision, uncemented fixation with better clinical recovery outcomes and equally low complications may be a more desirable option when considering revision TKA.
Round 2
Reviewer 2 Report
The authors modified many parts, and in these parts, the modifications were made to a certain degree of acceptance.
But what is the conclusion on the subject of this study? Both cemented and uncemented showed no difference in equivalent clinical results, revision rate, and stability. Is that the fact that the authors found?
In the now revised version of the paper, the conclusion is that uncemented was functionally better than cemented type. In this regard, the authors need to check once again the themes they want to find out when revising this paper.
In abstract,
As the authors indicate in the results, if you want to further analyze the difference in clinical results according to young age, please explain by adding this to the method.
I believe that the results and conclusions described by the authors are inconsistent.
line 45: With the worldwide increase in younger patients?
line 70-76: It is difficult for me to completely agree with these predictions and descriptions.
Table 1 needs to look good by adjusting the font size or the number of columns.
line 204: The term 'concluded' is inappropriate.
line 227-229: Is this a statistically proven result? I understood that there was no statistical difference in the results of the authors I read, am I wrong?
please rewrite the 'conclusion'.
the price of TKR might be different depending on the environment or medical system in each nations. I can not agree that mention of "not expensive"... I think this conclusion have to include the comparable outcomes, revision rate and stability of both style fixation methods.
Author Response
Response to Reviewer 2
Dear reviewer,
Sincerely thank you again for your constructive comments, in the process of replying to you, we also made our research more scientific and rigorous. Thank you very much for your comments and professional advice. These opinions help to improve academic rigor of our article. Based on your suggestion and request, we have made corrected modifications on the revised manuscript. We hope that our work can be improved again. Furthermore, we would like to show the details as follows:
Point 1: As the authors indicate in the results, if you want to further analyze the difference in clinical results according to young age, please explain by adding this to the method.
Response 1: Thank you for your kind advice. We then added this part in methods.
Point 2: line 45: With the worldwide increase in younger patients?
Response 2: This observation is correct. We have changed this sentence to “However, younger patients are undergoing elective knee replacement in higher numbers, there is a greater focus on improving functional recovery and quality of life after TKA [3, 4]” in line 48-51.
Point 3: line 70-76: It is difficult for me to completely agree with these predictions and descriptions.
Response 3: We have reorganized the paragraph structure and made it clearer and more accurate. We removed ambiguous descriptions and ensured that our descriptions and conclusions were based on published research results.
Point 4: Table 1 needs to look good by adjusting the font size or the number of columns.
Response 4: Thank you for your kind suggestion. We have changed the font size to 8 pounds to make it easier to read.
Point 5: line 204: The term 'concluded' is inappropriate.
Response 5: Thank you for your notice. We have corrected it in line 219.
Point 6: Is this a statistically proven result? I understood that there was no statistical difference in the results of the authors I read, am I wrong?
Response 6: We verified our results using the sensitivity analysis suggested in the Cochrane guidelines in order to test the stability of the conclusions. After removing Kim's study, the significance of the difference changed from p=0.01 to p=0.13. In a meta-analysis, the purpose of sensitivity analysis is to answer this question: "Are the findings robust to the decisions made in the process of obtaining them?" Therefore, we reviewed all the included studies and found that Kim's 2020 study did not have the potential for bias that would have significantly changed the conclusion and met the criteria for standard randomized controlled studies. No significant heterogeneity was found in the heterogeneity test of included studies (p=0.19). Similarly, there was no significant publication bias in the funnel plot. Moreover, the prosthesis used in Kim's study is one of the most used in the world, making it more representative. To sum up, we believe that Kim's study should not be excluded, and the final analysis results are reliable. The change of significance after sensitivity analysis can be attributed to the large sample size included in Kim's study in the result analysis, the large follow-up time span, different types of prostheses, etc., which were discussed in the limitations.
Point 7: Please rewrite the 'conclusion'. the price of TKR might be different depending on the environment or medical system in each nation. I cannot agree that mention of "not expensive"... I think this conclusion have to include the comparable outcomes, revision rate and stability of both style fixation methods.
Response 7: Thank you for your advice. According to the literature, the calculated cost of cemented TKA is $588 to $1043, depending on technique. The general increased charge for cementless TKA implants over cemented TKA implants was $366. However, in terms of time cost, mean operative time for cemented TKA was 11.6 minutes longer for cemented TKA than cementless TKA (93.7 minutes (SD 16.7) vs 82.1 minutes (SD 16.6); p = 0.001). Using a conservative published standard of $36 per minute for operating theatre time cost, the total time cost was $418 higher for cementing TKA. Overall costs and cost-effectiveness including both in-patient and out-patient costs at 1 year after cemented and uncemented knee replacement were comparable.
However, as you said, there are differences in costs across countries and regions. The literature we cited only includes data from some parts of the United States, so it is not applicable to all countries and region. But we think that with the increasing use of cementless prostheses, the cost will have less impact on surgical choice. We have rewritten this section to be more scientifically rigorous and to emphasize that this description applies only to specific regions.
Indeed, a summary of the results of all analyses is missing in the conclusion section. We have added this in the reworked manuscript.
In the latest revised manuscript, we have sorted out the research objectives and research conclusions, specifically clarified the conclusions of the research, and indicated our understanding of these statistical results in the discussion.
We would like to thank you again for taking the time to review our manuscript.